# Therapeutic Potential for Beta-3 Adrenoreceptor Agonists in Peripheral Arterial Disease and Diabetic Foot Ulcers

**DOI:** 10.3390/biomedicines11123187

**Published:** 2023-11-30

**Authors:** Cameron J. F. Evans, Sarah J. Glastras, Owen Tang, Gemma A. Figtree

**Affiliations:** 1Kolling Institute, University of Sydney, Sydney, NSW 2006, Australia; sarah.glastras@sydney.edu.au (S.J.G.); owen.tang@sydney.edu.au (O.T.); 2Faculty of Medicine and Health, University of Sydney, Sydney, NSW 2006, Australia; 3Department of Diabetes, Endocrinology & Metabolism, Royal North Shore Hospital, Northern Sydney Local Health District, Sydney, NSW 2065, Australia; 4Department of Cardiology, Royal North Shore Hospital, Northern Sydney Local Health District, Sydney, NSW 2065, Australia

**Keywords:** diabetes mellitus, peripheral arterial disease, wound, ulcers, oxidative stress, angiogenesis, nitric oxide, endothelial nitric oxide synthase, glutathionylation

## Abstract

Annually, peripheral arterial disease is estimated to cost over USD 21 billion and diabetic foot disease an estimated at USD 9–13 billion. Mirabegron is a TGA-approved beta-3 adrenoreceptor agonist, shown to be safe and effective in the treatment of overactive bladder syndrome by stimulating bladder smooth muscle relaxation. In this review, we discuss the potential use of beta-3 adrenoreceptor agonists as therapeutic agents repurposed for peripheral arterial disease and diabetic foot ulcers. The development of both conditions is underpinned by the upregulation of oxidative stress pathways and consequential inflammation and hypoxia. In oxidative stress, there is an imbalance of reactive oxygen species and nitric oxide. Endothelial nitric oxide synthase becomes uncoupled in disease states, producing superoxide and worsening oxidative stress. Agonist stimulation of the beta-3 adrenoreceptor recouples and activates endothelial nitric oxide synthase, increasing the production of nitric oxide. This reduces circulating reactive oxygen species, thus decreasing redox modification and dysregulation of cellular proteins, causing downstream smooth muscle relaxation, improved endothelial function and increased angiogenesis. These mechanisms lead to endothelial repair in peripheral arterial disease and an enhanced perfusion in hypoxic tissue, which will likely improve the healing of chronic ulcers.

## 1. Introduction

Peripheral artery disease (PAD) and diabetic mellitus are the two most common underlying causes for non-healing wounds in lower limbs [1], with more than USD 34 billion spent annually on PAD and diabetic foot ulcers (DFUs) in the US alone [2,3]. Globally, the prevalence of diabetes mellitus (DM) is expected to increase to 10.2% by 2030 [4]. DFUs are mostly attributed to ischemia from atherosclerotic PAD, which is often caused by poor glycaemic control which leads to endothelial oxidative stress and peripheral neuropathy [5]. Of the 537 million people living with DM in 2023, at least 1 in 5 will develop a DFU at some stage of their life [6,7]. Hence, facilitating safe and effective wound healing for patients with peripheral vascular disease-related ulcers (PVUs) and DFUs will have significant benefits in terms of reducing personal burdens and costly healthcare utilisations.

Current PAD treatments are limited to complex interventional procedures that revascularize tissues, and pharmaceuticals which slow the progression of atherosclerosis [8]. Other than vascular interventions, DFUs are treated with pharmaceuticals aimed at broadly improving metabolic control and dressings to keep wounds moist [9]. There is an unmet need for the medical treatment of the underlying sinister mechanisms of PAD, PVUs and DFUs.

Vascular ulcers develop due to the lack of oxygen and nutrients from persistent critical limb ischemia. This ischemia is caused by atherosclerotic lesions, arteriolar thickening and stenosis, amongst other vascular pathologies sometimes leading to necrosis and gangrene [10]. PVU, when combined with DM and/or peripheral neuropathy and/or trauma and/or infection, forms a DFU, which is clinically seen as a broken epidermis with a damaged dermis at the plantar areas of the foot [11,12,13]. Oxidative stress associated with hyperglycaemia and broader metabolic dysregulation directly contributes to high rates of PAD and poor wound healing in DM. Dysfunctional endothelium and inflammation within the vasculature contribute to both atherosclerosis and impaired microvascular perfusion [14]. In addition to reduced perfusion, angiogenesis is impaired in DM, limiting adaptive responses.

Nitric oxide (NO) is the smallest known biological signalling molecule [15]. It is an active endogenous regulator of inflammation, angiogenesis, leukocyte adhesion and vasodilation, and it is a direct ameliorator of oxidative stress through reactive oxygen species (ROS) scavenging [16,17]. Endothelial nitric oxide synthase (eNOS) is one of the three enzyme isoforms that produce NO. eNOS is found on the membrane of endothelial cells alongside the beta-3 adrenoreceptor (β3AR), L-type calcium channels and caveolin proteins [18,19]. Mirabegron is a small molecule β3AR agonist that is approved in Australia by the Therapeutic Goods Administration (TGA), and is currently used for the treatment of overactive bladder syndrome [20]. A more recent β3AR agonist, vibegron, is more efficacious and specific to the β3AR, and it is approved for use in overactive bladder syndrome in Japan and the US as of 2021 [21,22]. Agonist binding of the β3AR causes eNOS recoupling and activation facilitating NO synthesis as well as downstream vasodilation. The production of NO is key to restoring endothelial function in hyperglycaemia and oxidative stress [23].

The demonstrated clinical safety, efficacy and availability of β3AR agonist compounds make their repurposing for PAD and DFU possible. The β3AR is present in specific tissues including the endothelium, myocardium, smooth muscle and adipose tissues [24]. β3AR agonists reduce oxidative stress, increase perfusion, increase angiogenesis and restore endothelial cell function and vascular plasticity, which are all functionally relevant mechanisms in treating PAD and DFU [23,25]. Repurposing β3AR agonists as therapeutic agents for PAD and DFUs is a low-risk, non-invasive, cost-effective option that can complement current approaches to PAD and DFU, with the potential to improve tissue healing and reduce the demand for surgical interventions—the single biggest driver for non-traumatic major and minor amputations [6]. Here, we review the pharmacology and data supporting the potential benefits of stimulating the β3AR signalling pathways in patients with PAD and DFU.

## 2. Peripheral Artery Disease

Atherosclerotic PAD is a progressive disorder where arteries other than the coronary and cerebral arteries are occluded or stenosed by atherosclerotic plaques, leading to a reduction in the flow of oxygenated blood to the periphery, particularly the lower limbs where critical limb ischemia is the ultimate outcome [14,26,27].

The narrowing or occlusion of vessels in PAD leads to reduced perfusion and hypoxia, ultimately causing downstream tissue damage [1,6]. Microvascular disease (MVD) is highly correlated with PAD, although it is mechanistically independent of atherosclerosis. When combined, MVD and PAD carry a 23-fold risk of amputation. Reduced capillary density in the skin from a lack of angiogenesis causes cutaneous hypoxia, reducing wound healing capabilities and instigating ischemic ulcers [28]. Risk factors include DM, smoking, obesity, hypercholesterolaemia, hypertension and older age [5,29,30]. PAD is an independent risk factor of systolic heart failure and, importantly, the prevalence of heart failure doubles with PAD [31]. Concomitant congestive heart failure and venous pathology exacerbate poor peripheral perfusion, significantly slowing healing, increasing ischemic ulcer recurrence and increasing amputation incidence and morbidity [32]. A lack of perfusion limits tissue oxygenation, immune cell infiltration and debris clearing. Currently, angioplasty and bypass grafts are the mainstream interventions to treat PAD; these invasive methods potentiate heightened risk and cost, though strategies to reduce risk are constantly improving [8,33,34]. A recent retrospective US-based cohort study found that the average cost for angioplasty was USD 120,586 and USD 50,516 for an arterial stent. There was a high likelihood of repeat procedures within 2 years following endovascular procedures, drastically increasing costs, which highlights the lack of long-term efficacy of treatment [35]. Following vascular procedures, medical management includes statins, antiplatelets and antihypertensives, which are intended to avoid the progression or recurrence of stenosis, atherosclerotic plaque, and thrombi, to slow the progression of disease and reduce cardiovascular events and mortality [29,36]. However, none of these medications specifically improve ischemia, nor do they attempt to combat atherosclerotic plaque development and associated endothelial oxidative stress. Doing so will provide a cost-effective solution to surgical intervention and will reduce the overall burden on the public health system.

## 3. Peripheral Neuropathy

Peripheral neuropathy (PN) is considered a major neurovascular complication of DM. A long duration of DM and persistent hyperglycaemia increase PN. Glycation end products, associated oxidative stress and downstream inflammatory pathways lead to the degradation of peripheral nerve fibres, contributing to the dysfunction and denervation of sensory, motor and autonomic nerves within peripheral tissues [37]. Vasomotor paresis is the loss of the autonomic control of blood vessels, and it leads to the closure of certain vessels within the foot, reducing perfusion to nerves and exacerbating ischemia [38]. Clinical consequences include foot deformity, hammer toes and muscle atrophy in motor neuropathy; high plantar pressure, pain, burning and numbness in sensory neuropathy; and drying of the skin and vasomotor paresis in autonomic neuropathy (Table 1). Current treatment options for PN, beyond the medical management of PAD and glycaemic management for DM, vary in effectiveness and often involve a protracted period when ulceration is present (Table 1) [38,39].

## 4. Diabetic Foot Ulcers

A DFU is a significant and complex complication of DM, substantially increasing the likelihood of lower limb amputation and morbidity [36,40,41]. Ulceration develops from the multifaceted pathologies associated with DM, often following minor trauma combined with infection [6,38,42]. DFUs are almost always associated with the presence of PAD and/or PN and/or infection [8,13]. DFUs are exacerbated by elevated pressure at weight-bearing plantar areas—often unnoticed by the individual due to hypoesthesia associated with polyneuropathy [38]. Older people are more likely to have reduced skin integrity, defective immune response and impaired angiogenesis in response to injury. Skin with reduced integrity and flexibility cannot resist pressures of weightbearing, and wounds often fail to completely heal [43,44,45]. Attempts to treat DFU with systemic antibiotics are less effective due to the lack of perfusion to the infected area. Medical treatments to improve clinical outcomes of DFU currently focus on resolving underlying mechanisms, such as glycaemic management, angioplasty and dedicated dressings, to assist in wound healing. However, many fail to successfully treat ulceration without consequences [46].

Amputation is a costly treatment strategy for DFU, significantly reducing the quality of life and life expectancy as a trade-off for sepsis and death. In 2020, re-amputation still occurred in up to 45% of patients at 5 years post-op [47]. The combination of DM and PAD, along with aging, malnutrition, repeated infection and tobacco abuse, cause cutaneous wounds to be held in a state of heightened inflammation and oxidative stress, allowing the formation of a chronic ulcer [41,48,49]. These chronic ulcers can develop without the presence of DM. Thus, the combination of an aging population, and a major DM epidemic, will result in an increased prevalence of DFUs and their complication if medical interventions do not rapidly improve [50].

## 5. Impaired Wound Healing from Hypoxia and Oxidative Stress

DM and atherosclerotic disease are the two most common underlying causes for non-healing wounds in lower limbs [1]. The repair of damaged tissues involves four distinct phases: haemostasis, inflammation, proliferation and remodelling. Ulceration, either vascular or diabetic, occurs from the prolongation or cessation of phases of cutaneous wound healing. Angiogenic factors, inflammatory signals, growth factors and cytokines control the formation of extracellular matrix [ECM] and granulation tissues and then their closure and remodelling into functional tissue post-injury [51]. However, the pathophysiology associated with DM and PAD alter the regulation of angiogenesis, inflammation, cell migration and matrix deposition, which are all essential processes for wound healing [8,48]. Reduced angiogenesis leads to reduced wound nutrient supply and leads to cell death [1].

In homeostasis, tissues are well-oxygenated, and the balance of pro- and anti-angiogenic factors regulates blood vessel formation. When acute trauma occurs, these factors are released by local cells and the tissue quickly becomes hypoxic, causing angiogenesis, keratinocyte proliferation and the recruitment of platelets to form a clot, reducing blood loss and achieving haemostasis [52].

Under pathophysiological conditions, prolonged hypoxia reduces the expression of pro-angiogenic factors, reducing angiogenesis in chronic wounds and slowing healing [53]. This hypoxic environment also signals leukocytes to congregate at the site of injury, producing reactive oxygen species (ROS) to fight infection. ROS are a family of molecules with oxygen groups that have a higher oxidative potential than molecular oxygen (O_2_). Inflammatory cells, including leukocytes and macrophages, involved with wound healing use ROS for cell signalling and defence [11,51,54]. The balance of oxidants and antioxidants is known as redox homeostasis. Low concentrations of both ROS and NO are important for normal cellular functions. However, at moderate or high concentrations, both can be harmful, inactivating critical cellular processes [16]. Increased ROS production occurs from glycosylation end products in hyperglycaemia and mitochondrial dysfunction, leading to an imbalance in oxidants and antioxidants—a phenomenon known as oxidative stress, which causes oxidative damage to proteins, cells, and tissues [30,55,56,57]. Oxidative damage of the endothelium leads to impaired migration and angiogenesis, vasoconstriction and the development of atherosclerotic plaques facilitating ischemia from the termination of peripheral blood vessels, as seen in PAD [5,16,19,23,54,58]. Prolonged hypoxia and oxidative stress promote inflammatory cascades, increasing immune cell infiltration and local ROS concentrations. This worsens oxidative damage and holds the wound in a state of inflammation [11].

Ordinarily following inflammation, damaged tissues enter the proliferation phase, where fibroblasts and myofibroblasts rapidly multiply, producing collagen and ultimately forming granulation tissues with keratinocytes migrating to the wound. Oxidative stress together with aging and inflammation facilitate pre-mature cell senescence, leading to a hardening of the dermis and malformation of the epidermis [1,11,59,60]. Interestingly, there are cytokine and transcriptional profile differences in DFU and non-DFU. DFU displays higher pro-fibrotic and inflammatory markers and reduced anti-apoptotic markers compared to non-DFU [60]. Aside from their differences, chronic wounds are often a result of oxidative stress and hypoxia associated with vascular disease.

Targeting oxidative stress, angiogenesis and vascular tone may be the most direct ameliorator for chronic wound healing. Reperfusion of ischemic tissues and the reduction of oxidative stress will allow the tissues in wounds to receive nutrients and progress through wound healing phases normally, allowing for functional remodelling. NO can scavenge and reduce circulating concentrations of ROS, acting as an antioxidant and helping proteins, cells and tissues to function as normal [61].

## 6. Beta-3-Adrenoreceptor and Endothelial Signalling Pathways

The agonist stimulation of β3AR counteracts endothelial oxidative stress and limb ischemia by re-balancing redox conditions, increasing angiogenesis and facilitating NO-induced vasodilation, making it an interesting target for PAD and hypoxic wound healing [58]. The β3AR is the most recently discovered of the currently confirmed three adrenoreceptors (b1AR, β2AR and β3AR, respectively), and was first isolated in 1989 [62,63]. The β3AR is a G-protein coupled receptor (GPCR) expressed in the endothelium, myocardium, brain, retina, kidneys, bladder smooth muscle cells and adipose tissues [24]. GPCRs often contain phosphorylation sites for protein kinase A (PKA) and protein kinase C (PKC) or GPCR kinase (GRK), which, when bound, cause desensitisation and reduced receptor function. The β3AR is very resistant to the desensitisation of agonist-induced phosphorylation as it lacks PKA and PKC sites and only contains a single site for GRK [64,65,66]. This ability makes it an interesting therapeutic target, as it will continuously respond to targeted therapy [67]. Even with chronic activation with a mirabegron (agonist), the receptor shows little degradation, remains localised to the cell membrane, and maintains its function [64,68]. Conversely, β1- and β2ARs have numerous phosphorylation sites, facilitating their rapid catecholamine-induced desensitisation, rendering them ineffective long-term targets for therapy [63]. Importantly, the expression of the β3AR is shown to be upregulated in cardiovascular disease states and hypoxia [68,69,70]. The β3AR is colocalised with eNOS, nNOS, L-type calcium channels and caveolin (1 and 3) proteins in caveolae-enriched membranes of endothelial cells [65,71]. Caveolin 1 and 3 are proteins that form the caveolae of cell membranes [72]. Caveolin proteins inhibit eNOS by association, “clamping” and inactivating it. When a β3AR agonist is present, eNOS dissociates from caveolin and is activated with an influx of intracellular calcium from the activation of the L-type calcium channel via PKA, which also acts on eNOS itself (Figure 1) [73].

### 6.1. β3 AR Stimulation Activates eNOS

β3AR agonists stimulate the L-type Ca^2+^ channel current via the adenylate cyclase (AC), cyclic adenosine monophosphate (cAMP) and protein kinase A (PKA) pathway, leading to an increase in intracellular calcium. PKA also phosphorylates eNOS at Ser1177 and de-phosphorylates at Thr495, increasing Ca^2+^ sensitivity [74]. This activates eNOS, and is boosted by subsequent calmodulin binding after the Ca^2+^ influx as shown in Figure 1 [75]. The biochemical interactions that occur after mirabegron binds to endothelial β3AR are shown in Figure 1.

**Figure 1 biomedicines-11-03187-f001:**
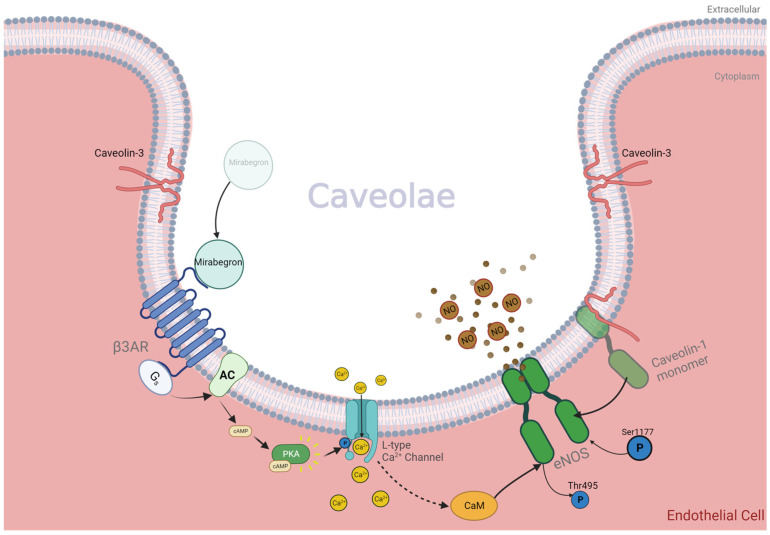
β3AR stimulated with mirabegron activates eNOS within caveolae of endothelial cells. Calcium influx is caused by phosphorylation of the L-type Ca^2+^ channel via the AC/PKA/cAMP pathway following β3AR-agonist stimulation [18,76]. Increased intracellular calcium strengthens the affinity of CaM to eNOS, subsequently decreasing the affinity of eNOS to inhibitory caveolin-1 [74,77]. Phosphorylation of eNOS at Ser1177, and dephosphorylation of inhibitory threonine 945 increases eNOS activity [25]. Thr495 is present on the CaM binding site, hence its inhibitory action on eNOS. Activated and coupled eNOS produces circulating NO which can diffuse to other tissues and scavenge ROS, reducing oxidative stress conditions [57]. Created with BioRender.com.

### 6.2. β3-Adrenoreceptor Stimulation Combats Oxidative Stress by Re-Coupling eNOS

As diabetes and PAD are both underpinned by oxidative stress, targeting nitric oxide production from eNOS with β3AR stimulation will reduce concentrations of ROS and combat oxidative stress directly. Ordinarily, eNOS exists as a homodimer. The structure and functionality of eNOS are modified by oxidative conditions. Depending on its conformation, eNOS can produce both NO and superoxide by being either coupled or uncoupled, respectively. In oxidative stress, eNOS is uncoupled, producing superoxide and worsening redox conditions [78]. Re-coupled eNOS after β3AR agonist stimulation improves redox conditions (Figure 2) through the cessation of superoxide production and the reinstated production of NO from eNOS, which can scavenge for ROS. Reduced oxidative stress results in the normalisation of inflammatory cascades and angiogenic gene expression profiles allow for controlled angiogenesis to take place and for the progression of wound healing phases to occur [11,79].

### 6.3. β3-Adrenoreceptor Stimulation Causes Vascular Smooth Muscle Cell Relaxation

PAD, PVU and DFU are all linked through ischemia and hypoxia. NO derived vasodilation in constricted vessels or vessels narrowed by plaque improves ischemia and resolves downstream hypoxia [58,85]. Vasodilation in macro- and micro-vessels is reduced in oxidative stress [42,86,87,88]. As well as combating an oxidative imbalance, eNOS-derived NO can diffuse into vascular smooth muscle cells, binding soluble guanylyl cyclase (sGC) and producing cyclic guanosine monophosphate (cGMP), which activates protein kinase G (PKG)—an enzyme responsible for the phosphorylation of vasodilator-stimulated phosphoprotein (VASP) at serine 239 [25,89,90,91] (Figure 3). This NO/sGC/cGMP/PKG leads to vasodilation, improving perfusion in hypoxic tissues.

With increased concentrations of circulating ROS, traditional methods for donating NO (either intravenously or using nitrate patches) are less effective. Recoupling and stimulating eNOS in diabetic tissue results in increased NO and hence decreases ROS within smooth muscle and endothelial cells, increasing NO sensitivity (particularly in DM) [86]. Functionality is restored by re-establishing redox homeostasis protein, allowing for the dilation of smooth muscles and increased angiogenesis, improving perfusion and cellular function in diseased tissues [58].

## 7. Nitric Oxide Synthases

There are three isoforms of nitric oxide synthase, each producing NO for different functions. They comprise neuronal (nNOS), inducible (iNOS) and endothelial (eNOS), each encoded by different genes on separate chromosomes [87]. nNOS regulates synaptic plasticity and blood pressure. NO from nNOS can also influence vascular tone in the periphery [15,93]. iNOS is involved in non-specific immunity, septic shock and acute inflammation. eNOS regulates vascular function and has an important role in vasoprotection, particularly in atherosclerosis [15,78]. Although eNOS and nNOS are coded on separate genes on separate chromosomes, they are both similarly activated by increased Ca^2+^ [87,94,95]. iNOS is functionally different, and is activated by stimuli such as bacterial lipopolysaccharides, cytokines and the acute phase of wounds [15,96]. This review focused on eNOS for its proven effect on vascular NO and relationship with the β3AR. However, the peripheral vasodilatory action of nNOS with β3AR agonist stimulation may also play a major role in the restoration of peripheral hypoxia [93,97]. Thus, further research into this relationship is needed, especially in the context of wound healing.

## 8. Future Directions and Opportunities for Repurposing β3-Adrenoreceptor Agonists

### 8.1. Mirabegron and Vibegron

Mirabegron and vibegron are small-molecule β3AR agonists approved for use in the treatment of overactive bladder syndrome [24]. Systemic mirabegron successfully causes smooth muscle relaxation in the detrusor muscle, allowing the bladder to fill to a greater volume, reducing the frequency and urgency of urination [20]. Angiogenic upregulation, redox rebalancing and vasodilation have been shown to occur in multiple studies in the presence of mirabegron due to eNOS recoupling, increased NO production and increased cGMP [58,84,98]. Adverse effects from systemic mirabegron include nasopharyngitis, dry mouth, cystitis and constipation [99].

Vibegron is the next generation β3AR agonist after mirabegron, approved in Japan in 2018 and the US in 2021, and is used in treating overactive bladder syndrome. Vibegron has no CYP2D6 or other major CYP enzyme inhibitory effect, reducing potential interactions with other CYP2D6 substrate drugs, such as metoprolol or desipramine [100]. Compared to mirabegron, it has a higher maximum response at the β3AR (99.2% vs. 80.4%), and fewer interactions with other adrenoreceptors, leading it to be thought of as safer, with a lesser likelihood for adverse reactions than mirabegron [21]. β3AR agonists effectively restore endothelial function, increase angiogenesis in diabetic ischemic tissue and improve redox conditions [58]. Treating PAD and DFU with repurposed clinically approved β3AR agonists may lead to better outcomes in patients with PAD, whilst also reducing associated medical costs.

### 8.2. Systemic Therapy

The systemic use of β3AR agonists facilitates angiogenesis within diabetic ischemic limbs and improves redox conditions, allowing for reperfusion of hypoxic tissues and better wound healing downstream [58]. However, mirabegron is shown to accelerate aortic atherosclerotic plaque development and instability in high-fat diet apolipoprotein E *(ApoE*) knockout mice (a pre-clinical model of atherosclerosis), which were administered a clinical dose orally (0.8mg/kg in C57BL/6 mice, equivalent to 50 mg in humans), likely due to increased circulating LDL and VLDL cholesterol remnants from adipose browning. Importantly, when *ApoE*^−/−^ and mitochondrial uncoupling protein 1 (UCP1) knockout mice were administered with mirabegron, their blood lipid profiles returned to normal and atherosclerotic plaque development was ceased, suggesting a maleficent role of UCP1 in plaque growth [101,102]. UCP1 is responsible for non-shivering thermogenesis from the browning of adipose tissues [103]. The potential vascular modifications from UCP1 lipolysis-dependent plaque growth and its relationship with β3AR stimulation presents an interesting opportunity for further studies of vascular effects mirabegron. In mice with a functional ApoE pathway, and normal levels of circulating lipids, the negative atherosclerotic effects of mirabegron are abolished [104]. Reassuringly, within human studies, oral mirabegron is not associated with a higher risk of cardiovascular disease [105].

### 8.3. Topical Therapy Potential in PAD and DFU

There are many advantages to topical administration. Firstly, increased local concentrations of the drug reduce systemic adverse reactions, and bypassing the gastrointestinal tract and first-pass metabolism in the liver prevents potential toxicities. Topically applied drugs can reach the intended tissue quickly and efficaciously, while providing a sustained and controlled dose [106].

Whilst topical applications present the shortest physical pathway to diseased tissues, this pathway involves penetrating the body’s strongest physical barrier, the stratum corneum. The stratum corneum is the outermost part of the skin comprising (mostly dead) specialised keratinocytes and oils. This layer is made even stronger with senescent tissue, often present in DFU. Once penetrated, the dermis contains an extensive microvascular network that can absorb and distribute a drug through the vasculature. The topical formulation must have a perfect balance of lipophilicity to permeate the stratum corneum whilst being able to further diffuse into the epidermis and dermis [107]. Therefore, topical therapy may better target the microvasculature and ischemic tissue in PAD and DFU [106,108]. Capillary and arteriole salvage from the activation of microvascular eNOS may be a mechanism in which ischemia is resolved. The restoration of cutaneous endothelium without the worsening of plaque development may be facilitated through the topical application of β3AR agonists.

Successful topical administration of β3AR agonists may reduce these adverse effects and enhance the targeted therapeutic effect on ischemic tissue in diabetic foot ulcers without drastic systemic side effects. Directly targeting endothelial cells and broadly treating the pathophysiology of hypoxic peripheral diabetic tissue presents a unique challenge that may be overcome by applying drugs topically. As senescent tissue presents a major barrier to topical drug delivery, the immediate topical application of drugs following debridement is likely to provide the best penetration into the dermis as drugs can affect endothelial cells and improve hypoxia.

### 8.4. Combination Therapies

Therapeutic-governing body-approved drugs that improve vascular function, metabolic processes and tissue viability can be rapidly translated for use in PVU and DFU in combination with β3AR agonists without time-consuming human safety trials. Both mirabegron and vibegron have this therapeutic potential for PAD and DFU. Repurposing and combining other approved drugs with β3AR agonists, potentially in a topical formulation, may magnify their therapeutic benefits. Combining drugs in this way allows for the tailoring of personalised medical care. Drugs can be added or removed from solutions to target specific pathological pathways, reducing potential adverse effects, and increasing the efficiency of successful wound closures. Each combination can be rapidly screened in vitro and ex vivo through efficient wound healing models and functional assays. Systemic delivery of drugs in PAD and DFU faces a unique hurdle, in that peripheral ischemia and vessel shunting reduce drug diffusion in affected tissues, rendering the medical treatment more difficult. This is particularly true in the case of treating infection in ischemic ulcers with oral antibiotics as therapeutic doses often struggle to reach areas of active infection due to peripheral arterial occlusions [109]. There is a potential therapeutic opportunity in antibiotics combined with β3AR agonists. This combination will allow for greater concentrations of antibiotics to circulate in peripheral tissues as a result of increased peripheral perfusion [109].

### 8.5. Challenges

Clinically, glycaemic management remains a high priority issue as hyperglycaemia is directly responsible for poor wound healing through numerous mechanisms [41]. Appropriate self-management of glycaemic levels is still the best strategy for improving outcome of DFU reducing PN and improving microvascular dysfunction [6,9]. Pre-clinical studies and translational drug research hurdles can be overcome using modern methods and a pragmatic design of in vitro, in vivo and ex vivo models. A comprehensive model for hyperglycaemia, ischemia and neuropathy, together with the various risk factors for PAD and diabetes in wound healing, remains a challenge for pre-clinical and translational studies. Humans are uniquely susceptible to vascular diseases to a far greater extent than other mammals [110]. Modelling a complex human-specific disease in different species continues to challenge basic science. Mice are a standard pre-clinical model used for wound healing studies due to their low cost and maintenance. However, the murine wound healing model has its downfalls. Wound closure in mice primarily occurs due to the contraction of local tissues, whereas it occurs due to cellular migration in humans. To improve the modelling of human healing, splints are used to hold excisional mouse wounds open, preventing contraction. These splints can be groomed off, causing a contraction of the wounded area and thus excluding the mouse from the study. Modelling drug efficacy in wound healing using two (or more) splinted wounds on a single mouse with a vehicle and drug wound, respectively, and this is also current standard practice [111]. This model reduces the number of mice needed for a study, which is advantageous from an ethical standpoint. However, there may be potential for transdermal systemic uptake of drug, reducing reliability of the model by improving healing rates on vehicle wounds. Solid controls must be in place to increase the significance of findings. Thus, further optimisations and validations are needed to yield a more robust in vivo wound healing model for improved clinical relevance.

Bringing new medicines to people in need is the most important aspect of drug research. However, clinical trials are a costly and time-consuming hurdle, necessary for the safety and efficacy screening of therapeutics before they reach the open market. Mirabegron and vibegron are shown to be safe for use in humans. Thus, their repurposing in PAD and DFU allows for the bypassing of the early stages in clinical trials, reducing the costs and time taken to obtain these medications for the people in need.

## 9. Conclusions

Increasing eNOS activity by stimulating the β3-adrenoreceptor resolves oxidative stress and hypoxia through redox re-balance and ischemic limb reperfusion. Approved β3-adrenoreceptor agonists, mirabegron and vibegron, provide viable options for therapeutic repurposing in peripheral arterial disease and diabetic foot ulcers. The data presented in this review supports their potential use in PAD and DFU and address their key mechanisms: reduction of oxidative stress, enhancement of blood flow through neovascularization as well as angiogenesis and promotion of vasodilation via NO production from eNOS.

## 10. Patents

G.A.F. reports patents on novel diagnostics and therapeutics for cardiovascular disease: US9,638,699B2 WO 2022/174309

## Figures and Tables

**Figure 2 biomedicines-11-03187-f002:**
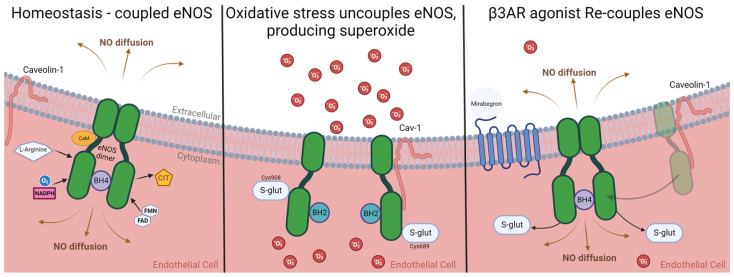
Mirabegron re-couples eNOS, after it is uncoupled by s-glutathionylation, association to caveolin-1 and increased the levels of BH2 from oxidative stress. (**Left**) Under favourable redox conditions, NO is produced by eNOS at the expense of the terminal guanidino nitrogen atoms of L-arginine and molecular oxygen, or reduced nicotinamide adenine dinucleotide phosphate (NADPH) as co-substrates with citrulline (CIT). Functional eNOS exists as a homodimer which is bound by haem and is essential for electron transfer between domains. Flavin adenine dinucleotide (FAD) and flavin mononucleotide (FMN) are co-factors. Calmodulin (CaM) is bound with increased Ca^2+^ concentration, facilitating electron transfer from NADPH. The binding of (6R-)5,6,7,8-tetrahydrobiopterin (BH4), through interactions with ZnS_4_, facilitates and stabilises the dimerization of the two eNOS monomers and increases the enzyme’s affinity to L-arginine. BH4 is essential for NO synthesis, when BH4 is depleted (often caused by oxidation and downregulation from hyperglycaemia in DM) or when levels of 7,8-dihydrobiopterin (BH2) are increased. (**Centre**) As oxidative stress conditions increase, eNOS is proportionally uncoupled by association with caveolin proteins (Cav-1) and S-glutathionylation at Cys689 and Cys908. In the uncoupled state, eNOS produces superoxide, which is facilitated by BH2. (**Right**) eNOS is re-coupled from β3AR stimulation, dissociated from caveolin and “un-glutathionylated”, allowing the enzyme to form a homodimer in the presence of BH4. Activated and re-coupled eNOS produces NO instead of superoxide [15,19,74,75,78,80,81,82,83,84]. Created with BioRender.com.

**Figure 3 biomedicines-11-03187-f003:**
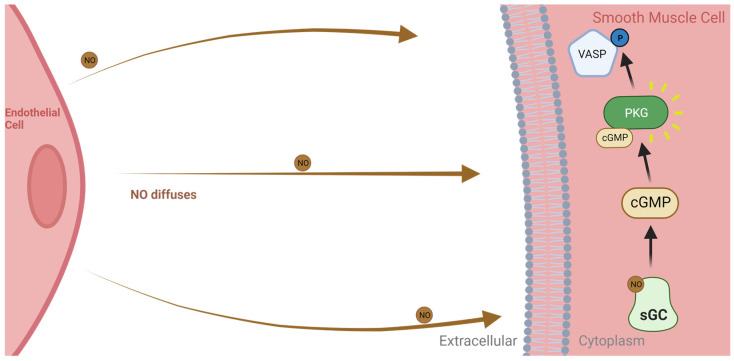
Nitric oxide from eNOS causes vasodilation in smooth muscle cells. NO diffuses into cells and interacts with sGC, creating cGMP and activating PKG, which phosphorylates vasodilator-stimulated phosphoprotein (VASP) and results in vasodilation of smooth muscle and endothelial cells [92]. Created with BioRender.com.

**Table 1 biomedicines-11-03187-t001:** Each type of peripheral neuropathy and its symptoms, causes and treatments.

Type ofNeuropathy	Sensory	Motor	Autonomic	Polyneuropathy
Symptoms	Burning	Atrophy	Vasomotor paresis	Unequal foot load
Paraesthesia	Hammer toe	Cracking of skin	Poor gait
Pain	Loss of reflex	Medial arterial sclerosis	Foot deformity
Numbness (chronic)	Cramps	Charcot’s foot	Increased forefoot pressure
CurrentTreatment	Pain relief	Removal of necrotic tissue	Angioplasty	Amputation
Offloading	Moist dressings	Revascularisation	Debridement
Skin grafting and substitution	Negative pressure wound therapy	Amputation

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
