# Peer review of "Therapeutic Potential for Beta-3 Adrenoreceptor Agonists in Peripheral Arterial Disease and Diabetic Foot Ulcers"

_biomedicines, 2023, doi:10.3390/biomedicines11123187_

Round 1
Reviewer 1 Report
Comments and Suggestions for Authors
Dear authors
This is a good review study. There are various similar studies in the filed.
The writing is acceptable.
The references are suitable and updated.
I observed that abstract states mechanisms but not potential candidate agents.
I also propose to add a sub-section regarding potential candidates targeting β(3)-adrenoceptor. Also, this sub-section can have a figure exhibiting sites of acting/targeting and cellular signals.
Also in the conclusion, potential candidates have not been mentioned.
No other concerns were observed.
Best regards
Author Response
Dear reviewer,
Thank you for your time and comments on this review, all edits to the manuscript are highlighted in red within the resubmitted document.
The following is our response to your feedback:
- "The abstract states mechanisms but not potential candidates", Mirabegron is mentioned in line 13 and is the main focus for this review other than vibegron.
- "add a sub-section regarding potential candidates targeting β(3)-adrenoceptor...have a figure exhibiting sites of acting/targeting and cellular signals.", Subsection 8.1 - Mirabegron and vibegron is a subsection that describes mirabegron and vibegron. The binding action of these drugs is out of the scope of this review as we focus on the therapeutic potential within PAD and DFU. It is also a topic that has been well covered in literature and the biochemistry of the beta-3 ar and its agonists is referenced throughout this review.
- "Also in the conclusion, potential candidates have not been mentioned." Thank you for noticing this point, we have added this to the conclusion.
Thank you again for your time reviewing this manuscript.
Reviewer 2 Report
Comments and Suggestions for Authors
Interesting review on one of the most critical topic in Angiology and Vascular Surgery. Following are my suggestions to improve the manuscript.
Lines 13 & 63: explain the meaning of TGA.
Line 82: the acronym “PAD” has been already introduced in line 31.
Lines 82-5: please, add here this recent reference which is very specific to the point you are dealing with: Martelli, E.; Enea, I.; Zamboni, M.; Federici, M.; Bracale, U.M.; Sangiorgi, G.; Martelli, A.R.; Messina, T.; Settembrini, A.M. Focus on the Most Common Paucisymptomatic Vasculopathic Population, from Diagnosis to Secondary Prevention of Complications. Diagnostics (Basel), 2023, 13(14), 2356, doi: 10.3390/diagnostics13142356.
Lines 98-9: “both being expensive, invasive, and risky”. I would eliminate this statement: none of these adjectives is correct to define angioplasty or peripheral bypass graft. Each case can be so different from the others that the variables involved are too many.
Line 150: DM, not DMs.
References should be reported according to Biomedicines requirements. For instance:
- the first names of the authors in reference # 6 should be as initials;
- synthetize the name of the journal in references # 13, 14, 29, 30, 33, 34, 36, 38, 40, 47, 48, 50, 60, 71, 73, 77, 82, 83, 86, 88, 89, 91, 93, 97, 98, 100, 104-6;
- reference # 94 is not complete.
- the name of the journal is duplicated in reference # 95.
Author Response
Dear Reviewer,
Thank you for your helpful feedback on this review. All modifications have been highlighted in red for ease of reading within the resubmission.
- Meaning of TGA has been defined in the text.
- PAD acronym has been fixed, thank you.
- Thank you for the suggested reference, we agree that the point did need further reading. However, we have chosen a reference which we believe is more aligned to the focus of the review, diabetes and PAD.
- The wording of “both being expensive, invasive, and risky” has been changed to more suitably describe the risks and costs involved with interventional procedures. We believe it is important to highlight that there are risks and costs involved with current therapies so we decided not to eliminate the statement.
- DM fixed
- References #6, #94, #95 have been fixed.
- "synthetize the name of the journal in references..." we are unsure what is meant by this... perhaps you could further explain this point?
Thank you again for your time reviewing this article.
Reviewer 3 Report
Comments and Suggestions for Authors
The paper review the pharmacology and data supporting the potential benefits of stimulating the Beta 3AR signalling pathaways in patients with PAD and DFU.
It is a new area that authors show innovate rewiew for therapeutic repurposing in PAD and DFU.
The paper merits for publication just one question if possible that authors can add in challenges paragraph if there studies about clinical trails using Mirabegron, Vibegron or other products.
Author Response
Dear reviewer,
Thank you for your time spent reviewing this paper.
Clinical trial challenges comments have been added to the review, as requested, highlighted in red.
Thank you again for your feedback.